# Reliable Model Watermarking: Defending against Theft without Compromising on Evasion

Hongyu Zhu*
Southeast University
Nanjing, China
213200421@seu.edu.cn

Sichu Liang*
Southeast University
Nanjing, China
213200709@seu.edu.cn

Wentao Hu
Southeast University
Nanjing, China
213201262@seu.edu.cn

Li Fangqi
Shanghai Jiao Tong University
Shanghai, China
solour_lfq@sjtu.edu.cn

Ju Jia†
Southeast University
Nanjing, China
jiaju@seu.edu.cn

Shi-Lin Wang
Shanghai Jiao Tong University
Shanghai, China
wsl@sjtu.edu.cn

## Abstract

With the rise of Machine Learning as a Service (MLaaS) platforms, safeguarding the intellectual property of deep learning models is becoming paramount. Among various protective measures, trigger set watermarking has emerged as a flexible and effective strategy for preventing unauthorized model distribution. However, this paper identifies an inherent flaw in the current paradigm of trigger set watermarking: evasion adversaries can readily exploit the shortcuts created by models memorizing watermark samples that deviate from the main task distribution, significantly impairing their generalization in adversarial settings. To counteract this, we leverage diffusion models to synthesize unrestricted adversarial examples as trigger sets. By learning the model to accurately recognize them, unique watermark behaviors are promoted through knowledge injection rather than error memorization, thus avoiding exploitable shortcuts. Furthermore, we uncover that the resistance of current trigger set watermarking against removal attacks primarily relies on significantly damaging the decision boundaries during embedding, intertwining unremovability with adverse impacts. By optimizing the knowledge transfer properties of protected models, our approach conveys watermark behaviors to extraction surrogates without aggressive decision boundary perturbation. Experimental results on CIFAR-10/100 and Imagenette datasets demonstrate the effectiveness of our method, showing not only improved robustness against evasion adversaries but also superior resistance to watermark removal attacks compared to state-of-the-art solutions.

## CCS Concepts

• **Security and privacy** → *Human and societal aspects of security and privacy*; • **Information systems** → *Multimedia information systems*; • **Computing methodologies** → *Artificial intelligence.*

*Equal contribution.
†Corresponding author.

## Keywords

model watermarking, intellectual property protection, evasion adversary, stealing adversary

**ACM Reference Format:**
Hongyu Zhu, Sichu Liang, Wentao Hu, Li Fangqi, Ju Jia, and Shi-Lin Wang. 2024. Reliable Model Watermarking: Defending against Theft without Compromising on Evasion. In *Proceedings of the 32nd ACM International Conference on Multimedia (MM '24), October 28-November 1, 2024, Melbourne, VIC, Australia.* ACM, New York, NY, USA, 10 pages. https://doi.org/10.1145/3664647.3681610

## 1 Introduction

Over the past decade, significant advancements in deep learning have led to its widespread application in fields such as computer vision, natural language processing, and speech recognition [37]. Recent breakthroughs in Large Language Models (LLMs), such as ChatGPT, underscore the potential strides toward General Artificial Intelligence (AGI) [58]. These advancements have enabled companies such as OpenAI and Google to offer Machine Learning as a Service (MLaaS) via APIs, transforming sophisticated models into paid services accessible to the public. However, this also presents opportunities for adversaries to steal models [51], seeking to produce knockoffs and establish pirated API services for profit. The stolen victims involve significant investment including data annotation, expert knowledge and computational resources. For instance, training GPT-3 incurs a cost of approximately 12 million USD [4]. Therefore, model thefts severely damage the intellectual property and legitimate rights of the model owners.

Model stealing typically occurs via two approaches. First, adversaries may directly steal the parameters, highlighted by the leak of Facebook's LLaMa model [25, 60]. Second, attackers might prepare unlabeled data to query the target API, employing the probability labels to distill knowledge into a surrogate model. Known as model extraction or functionality-stealing attacks [52], this strategy exploits legitimate black-box access, making it challenging for owners to distinguish between benign users and potential thieves.

Preventing model theft at source is exceedingly difficult. Inspired by digital watermarking used to protect multimedia content [32], model watermarking is proposed as copyright identifiers to determine if a suspect model is a knockoff [76]. *White-box* watermarks [11] directly embed secret patterns into parameters, but require access to the suspect model parameters during verification, which

may not be feasible in real-world scenarios. Thus, *black-box* watermarks [1] have emerged as the predominant approach, requiring only access to the model's outputs for the trigger sets. Typically, model owner generates a set of secret watermark samples with deliberately incorrect labels, using techniques like backdoor injection to ensure the model memorizes this trigger set. If a suspect model produces the predefined labels for the trigger set with a high probability, it is identified as a copy of the protected model.

Memorizing a trigger set that deviates from the main task distribution will inevitably impair the generalization performance. However, due to benign overfitting [53], if the size of the trigger set is maintained within capacity limits [39], poisoning-style watermark embedding does not significantly degrade performance on standard testing benchmarks [1]. Consequently, the adverse effects of trigger set watermarks are often underestimated [3, 31, 33, 46, 85].

In this work, we identify that all poisoning-style watermarks, even those crafted with random label noise trigger sets, embed shortcuts into the protected model [20]. Evasion adversaries can readily exploit these shortcuts, employing efficient optimization frameworks to achieve significantly higher attack success rates than unwatermarked models. Hence, while designed to protect intellectual property from theft, poisoning-style watermarking inadvertently introduces severe vulnerabilities to evasion attacks. Furthermore, we identify a robustness pitfall phenomenon: current watermarks aggressively disrupt the decision boundary, generating misclassifications around watermark samples to achieve resistance against removal attacks. This trivial mechanism inadvertently entangles watermark unremovability with its adverse effects. To ensure effective watermark verification, representational capacity of the model must be sacrificed to focus on watermark behavior, laying severe risks for generalization in adversarial scenarios.

In response to the vulnerabilities identified in this paper, we revisit the pipeline of trigger set watermarks, proposing a reliable algorithm that resists removal attacks without increasing evasion risks. Instead of error memorization, knowledge injection is utilized to foster unique watermark behaviors. Specifically, we leverage diffusion models to generate Unrestricted Adversarial Examples (UAEs) from random noise, ensuring diversity, hardness, and fidelity in creating a versatile trigger set. Building on this foundation, we identify optimization difficulties in replicating the protected model as the primary reason for watermarks failing to survive extraction. Thus, we enhance the knowledge transfer properties of the watermarked model during embedding, learning it as a "friendly teacher" to effectively guide the surrogate model in acquiring watermark behavior from a limited query set, without relying on any robustness pitfall phenomenon. The whole pipeline is shown in Figure 1.

Our contributions are summarized as follows:

(1) We reveal that all poisoning-style watermarks embed exploitable shortcuts into the model and provide a detailed assessment of the evasion vulnerabilities introduced.

(2) We propose utilizing diffusion models to synthesize UAEs as the trigger set, devising effective generation algorithms that enable harmless watermarking via knowledge injection.

(3) We identify the robustness pitfall that brings contradictions between generalization and watermark unremovability. We reconceptualize the embedding process by focusing on knowledge transfer properties of the protected model.

(4) Integrating analyses and designs above, we propose the first reliable watermarking algorithm that demonstrates improved evasion robustness and surpasses current state-of-the-art methods in watermark unremovability.

## 2 Related Work

### 2.1 Model Stealing and Watermarking

With the widespread application of deep learning, associated models have increasingly become targets for theft. *Parameters Stealing* directly acquiring models through cyberattacks and social engineering [65], or reversing model parameters through side channels [60, 82]. *Functionality Stealing* utilize unlabeled data to query the target API and leverage the soft labels to train a surrogate model via maximum likelihood estimation, approximating the functionality of the stolen victim on the target task [52, 56, 75, 84]. Based on the principles of knowledge distillation [26, 78], *functionality stealing* is scalable to large language models (LLMs) [35, 73], and is highly feasible in open world scenarios.

To effectively counteract complex and varied thefts, embedding digital watermarks into protected models is a straightforward and effective strategy [32]. This allows the tracking of infringing actions by transferring watermarks to knockoffs obtained by adversaries. *Parameter embedding* watermarks [11, 76] directly implant secret patterns into model parameters, but the verification requires white-box access, which can be denied by the model owner. Furthermore, these patterns are often fragile [40, 54]. Hence, *trigger set* watermarks [1], which only require black-box access to the suspect model have become the most popular method. Poisoning-style watermarks embed a trigger set that deviates from the main task distribution, exploiting unique behaviors on the trigger set during verification to assert model copyright [1, 3, 31, 33, 46, 85]. Alternatively, watermark behaviors can be reflected by directly modifying the API's probability outputs [70, 79], though it may not be secure against parameter stealing. Notably, non-intrusive fingerprints that characterize decision boundaries are feasible [45], yet are easier to remove [33, 79, 83] and prone to false alarms [69].

### 2.2 Watermark Removal and Countermeasures

In the ongoing challenge for intellectual property protection, attackers strive to remove watermarks from stolen models while preserving their generalization performance. *Model modification* [44] involve slight alterations such as fine-tuning [43] or pruning [15] to make the model forget the trigger set. *Input preprocessing* [44] transforms input samples [23] to evade from watermark triggering, or employs anomaly detection [41] to filter out suspicious watermark inputs. *Model extraction* [44], primarily used for functionality stealing, also effectively removes watermarks since the trigger set usually does not appear in the extraction query set, complicating the transfer of watermark behaviors during the process of functionality approximating .

In response to removal attempts, robust embedding algorithms are proposed to fortify watermark behavior. Entangled Watermark Embedding (EWE) [31] tightly couples watermark samples with the task distribution through soft nearest neighbor loss regularization. Random smoothing (RS) in parameter space [3] provides certification of unremovability against minor parameter changes. Margin-based watermarking (MBW) [33] increases the margin between

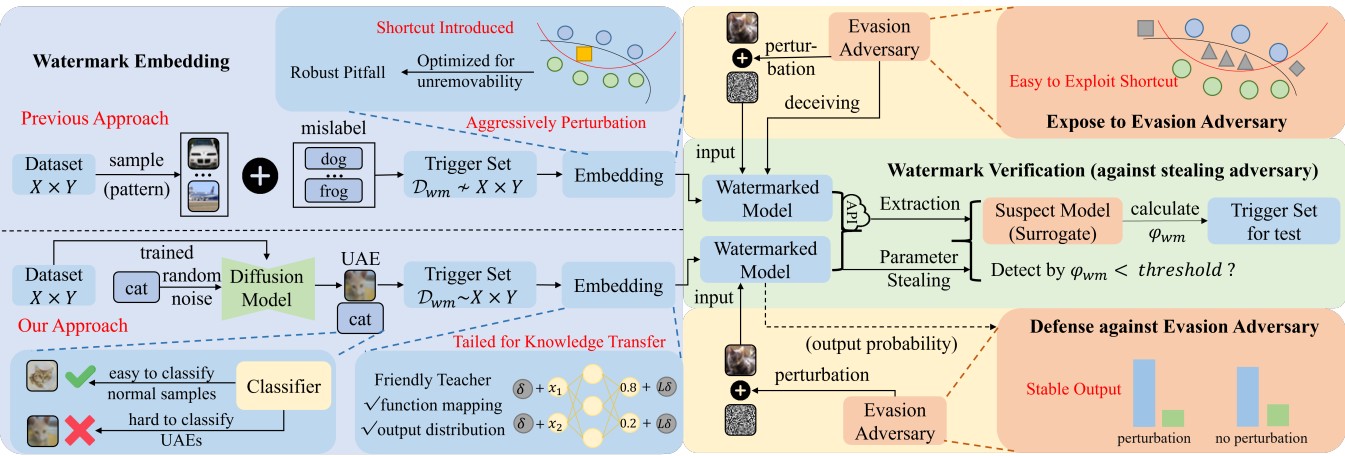

Figure 1: Comparison of Watermark Embedding Pipeline in this Work and Previous Works.

watermark samples and the decision boundary via adversarial training, increasing the likelihood of watermark retention during model extraction. MEA-Defender [46] employs mixed samples from different source classes as watermarks, binding the trigger set to main task distribution by minimizing the Kullback-Leibler divergence. Overall, robust watermarking algorithms embed the trigger set deeper within the original distribution, significantly altering the decision boundary to boost resilience of the watermarks.

## 3 Exploiting Watermark Shortcut

Recent studies have shown that various failures in machine learning systems can be attributed to shortcut learning—models prefer easy-to-learn patterns over complex, generalizable ones [20]. During standard training, models seek to map inputs to outputs via empirical risk minimization (ERM), a process prone to capturing superficial correlations [62]. While such models excel in controlled test environments, their performance deteriorates in the real world characterized by distribution shifts or adversarial attacks, stemming from the reliance on domain-specific or non-robust features [19, 29]. Shortcuts often arise from dataset biases or learning dynamics of ERM optimizers [74]. Yet, we reveal that poisoning-style watermarking deliberately introduces shortcuts into models, creating hidden pathways adversaries can exploit, significantly amplifying vulnerabilities under evasion attacks.

First, we formally define the process of trigger set watermark embedding to expose **why shortcuts exist**.

### 3.1 Formulation of Trigger Set Watermarking

In a $K$-class classification problem, a model $f$ parameterized by $\theta$, maps inputs $X \in \{0, 1, ..., 255\}^{C \times W \times H}$ to labels $Y \in \{1, ..., K\}$. Model owner generates a training dataset $\mathcal{D}_{tr} \in X \times Y$ from the underlying distribution $P(X, Y)$, minimizing $\mathcal{L}(f(x), y)$ over $D_{tr}$ to learn the mapping. Model performance is evaluated on a validation set $D_{val}$ with accuracy $score(\varphi_{acc}(f, \mathcal{D}_{val}) = \frac{1}{|\mathcal{D}_{val}|} \sum \mathbb{I}(f(x) = y))$. For trigger set watermark, a secret dataset $\mathcal{D}_{wm}$ with unique mappings $x_{wm} \rightarrow y_{wm}$ not present in $P(X, Y)$ is selected by the owner. Training on $\mathcal{D}_{wm}$ endows the watermarked model $f_{wm}$ with the capability to generate predetermined outputs on $x_{wm}$. Analog to its counterpart, watermark accuracy $\varphi_{wm}(f_{wm}, \mathcal{D}_{wm}) = \frac{1}{|\mathcal{D}_{wm}|} \sum \mathbb{I}(f_{wm}(x_{wm}) = y_{wm})$ measures $f_{wm}$'s adherence to $\mathcal{D}_{wm}$. The selection of triggers set can be categorized into two types:

**Pattern-based watermark.** This backdoor style[80] approach[1, 3, 31, 61, 85] enables the model to map samples with a specific trigger pattern to a predetermined target class. Watermark samples are created by overlaying the trigger pattern $\delta \in \{0, 1, ..., 255\}^{C \times W \times H}$ with mask $m \in \{0, 1\}^{C \times W \times H}$ onto samples $x_s$ from the source class $s$, resulting in $x_{wm} = x_s \odot (1 - m) + \delta \odot m$. For watermark embedding, the model $f_{wm}$ is adjusted to classify any sample with the trigger pattern into target class $t$ through optimization $\min_{\theta_{wm}} \mathcal{L}(f_{wm}(x_{wm}) = y_t)$. Given the secrecy of $\delta$, $m$, $s$, and $t$, knowledge of this backdoor serves as the proof of ownership.

**Pattern-free watermark.** Pattern-based watermark is challenged by the continuous evolution of backdoor defenses [21]. Recent strategies advocate for the use of any sample deviates from the main distribution $P(X, Y)$ as the trigger set $\mathcal{D}_{wm}$, eliminating the reliance on a fixed trigger pattern. Pattern-free watermarks could be mixed samples from two classes [46] or purely label noises [33].

Trigger set watermarking hinges on overfitting to a unique set distinct from the main distribution $P(X, Y)$, leading to shortcuts from $x_{wm}$ to $y_{wm}$. Although the potential harm of these shortcuts is often overlooked in literature, we then introduce straightforward optimization frameworks demonstrating **how** evasion adversaries can effortlessly **exploit these shortcuts**, significantly compromising the model's performance in adversarial settings.

### 3.2 Evasion Attacks via Watermark Shortcut

In evasion attacks, adversaries craft a distortion $\delta$ within an $l_p$ norm boundary, changing prediction of $f_{wm}$ when added to an original sample $x$ so that $f_{wm}(x + \delta) \neq f_{wm}(x)$. Equation 1 aims to find a perturbation $\delta$ that leads $f_{wm}$ to incorrectly classify instances from class $s$ to $t$, analog to the standard trigger inversion framework [74, 77]. Though trigger inversion has been evaluated in watermarking to neutralize suspicious inputs [31, 46, 77], it is dismissed as ineffective due to mismatches between recovered and actual patterns. Nonetheless, minimal modifications can activate backdoors [55], indicating exact pattern matching is unnecessary for evasion. By blindly optimizing $\delta$, we demonstrate the feasibility of deceiving $f_{wm}$.

Additionally, with source and target classes hidden, a brute-force search across class pairs is computationally challenging [77]. Thus, we investigate a universal attack as shown in Equation 2, aiming to maximize classification errors for global $\delta$.

Successfully solving Equations 1 and 2 reveals that while watermarked models maintain high accuracy on natural samples, the embedded shortcuts greatly increase susceptibility to evasion attacks. Moreover, secrecy of the trigger pattern in pattern-based watermarks does not safeguard against adversarial exploitation.

$$\delta^* = \arg \min_{\|\delta\|_p \leqslant \varepsilon} \mathcal{L}\left(f_{wm}\left(x_s + \delta\right), y_t\right), \forall (x_s, s) \in \mathcal{D}_{tr}. \quad (1)$$

$$\delta^* = \arg \min_{\|\delta\|_p \leqslant \varepsilon} -\mathcal{L}\left(f_{wm}(x + \delta), y\right), \forall (x, y) \in \mathcal{D}_{tr}. \quad (2)$$

The question arises: can forgoing patterns in watermarking prevent associated vulnerabilities? Theory indicates that memorizing any noisy data undermines robustness to adversarial attacks [53]. Notably, risk from purely random label noise can rival that of the most sophisticated poisoning attacks [53]. To illustrate this empirically, we conduct adversarial attacks on $f_{wm}$ guided by an optimization goal akin to Equation 2. Here, $\delta$ is customized for specific samples rather than a universal pattern, showing higher flexibility.

For noise label watermarks, beyond empirically exploring the increased adversarial risk, we delve into model vulnerability, explicitly linking decreased robustness to shortcut exploitation. Recent studies illustrate that backdoors can be implanted by merely altering labels [30]. Exploring the reverse scenario, we investigate whether a trigger pattern can be extracted from arbitrarily mislabeled samples to deceive the model. Algorithm 1 establishes a stochastic gradient approach (NLTI) to solve this problem. It operates on noise set samples in $X_{noise}$ where $y_{noise} = f_{wm}(x_{noise}) \neq y_{ori}$. The intuition is to assume a pattern $\delta$ has already been added to all samples in $X_{noise}$, causing the model to learn the corresponding incorrect labels in a backdoor injection manner. The algorithm aims to find this pattern such that all $x - \delta$ can be correctly classified:

$$\delta^* = \arg \min_{\|\delta\|_p \leqslant \varepsilon} \mathcal{L}\left(f_{wm}(x - \delta), y\right), \quad \forall x \in X_{noise}, y \in Y_{ori}. \quad (3)$$

---

**Algorithm 1** Noise Label Trigger Inversion.

---

1: **Input:** sample set $X_{noise}$, ground truth set $Y_{ori}$, watermark model $M$ and its wrong prediction $Y_{noise} = \{f_{wm}(x) \mid x \in X_{noise}\}$, learning rate $\alpha$, schedule $S$, perturbation bound $\varepsilon$.
2: **Output:** trigger pattern $\delta$.
3: **for** epoch $= 1 \ldots N$ **do**
4:     **for all** $(x, y, y') \in (X_{noise}, Y_{ori}, Y_{noise})$ **do**
5:         $\delta \leftarrow \delta - \alpha \cdot \nabla_\delta \mathcal{L}(x + \delta, y, y')$.
6:         Project $\delta$ to the $l_p$ ball with bound $\varepsilon$.
7:     **end for**
8:     Update $\alpha$ with learning rate schedule $S$.
9: **end for**

---

The pattern $\delta$ represents the source of misclassifications from the noise set, reflecting the shortcut created by noise memorization. Thus, applying $\delta$ to new samples is likely to confuse the model $(f_{wm}(x + \delta) \neq f(x))$. During optimization, the logits loss [5, 9] shown in Equation 4 is applied to enhances the correct class logits $z_y$ while pushing outputs away from the noise labels. Additionally, the learning rate is updated according to a schedule to balancie exploration and exploitation [9].

$$\mathcal{L}(x + \delta, y, y') = -z_y + z_{y'}. \quad (4)$$

NLTI mirrors the approach of universal adversarial perturbations (UAP) [64]. However, a key difference is that NLTI focuses on correcting the model's responses to de-noised samples while inducing errors in new samples with added $\delta$. Ultimately, the optimization

strategies offer an optimistic upper bound on the robustness against evasion. With persistent shortcuts, the exploitation capabilities of evasion adversaries will escalate with advancing attack techniques, progressively undermining generalization of watermarked models.

## 4 Towards Harmless Watermarking

Even memorizing random label noise in poison-style watermarks inevitably introduces exploitable shortcuts to the model [53]. Thus, methods relying on **misclassification of specific samples** for ownership verification are fundamentally flawed. Safe trigger-set watermarking is possible only by ensuring **unique correct responses to specific samples**. Building on this principle, we propose a harmless watermarking scheme focusing on trigger-set generation, watermark embedding and watermark verification.

### 4.1 Trigger-set Generation

*4.1.1 Motivation.* While the optimization in modern deep networks produces diverse solutions [28], predictions by various models on in-distribution samples often converge. The challenge lies in identifying a set of samples that prompts models to exhibit uniquely correct behavior. First, we summarize the conditions that samples used to construct a harmless trigger set should satisfy:

(1) **Clarity**: Possess clear semantics with a definite correct label;
(2) **Hardness**: Sufficiently difficult to ensure that correct predictions highlight distinct capabilities;
(3) **Stealth**: Closely resemble the original data distribution to bypass anomaly detection;
(4) **Resilience**: Maintain uniqueness against adaptive attacks.

Straightforward choices such as out-of-distribution and adversarial examples are challenging enough but either detectable [41] or highly fragile [8, 23]. In this paper, we introduce Unrestricted Adversarial Examples (UAEs) [66] as the trigger set. Free from $l_p$ norms constraints, UAEs provide superior flexibility in fooling classifiers and higher resistance to defenses [67]. We further design a pipeline for synthesizing UAEs via diffusion models [27], with powerful distribution prior [38] ensuring UAEs to be stealth [6, 81]. Moreover, diffusion models can synthesize infinitely realistic and diverse samples from random noise, producing elusive trigger sets.

*4.1.2 UAE Generation.* Diffusion models introduce Gaussian noise to samples in the forward process, leading to an isotropic Gaussian distribution. Conversely, the reverse process reconstruct samples from Gaussian noise [27]. This is achieved by training a denoising model $R_\Phi(x_\tau, \tau)$ progressively removes noise during the $T$-step schedule, ultimately recovering sample $x_0$ as shown in Equation 5:

$$p(x_{0:T}) = p(x_T) \prod_{\tau=1}^{T} p_\Phi\left(x_{\tau-1} \mid x_\tau\right). \quad (5)$$

Here, $x_T$ is the initial Gaussian seed for the reverse process, guided by the denoising model $R_\Phi(x_\tau, \tau)$ as $p_\Phi(x_{\tau-1} \mid x_\tau)$. Introducing a conditioning variable $c$ allows for a conditional diffusion model $R_\Phi(x_\tau, \tau, c)$ [13]. Class-conditioned diffusion provides distribution priors to align generated samples with class semantics, guaranteeing UAEs bear unambiguous labels. Synthesizing UAEs involves steering the generation process to fool classifiers while preserving class semantics. The adversarial nature emerges in three stages: seed selection, denoising trajectory, and final adjustment. We develop efficient methods for crafting UAEs at each stage:

**Adversarial Warm-up.** The objective is to subtly modify the seed $x_T$ such that its denoised result can deceive the classifier ($f(x_0) \neq c$). EvoSeed [34] uses a genetic algorithm to add adversarial perturbations to $x_T$, yet it demands thousands of iterations to converge. Therefore, we propose directly maximizing the loss of the classifier $f$ mapping $x_0$ to class $c$ through Equation 6:

$$x_T^{\alpha+1} = P_{Bp(x_T,\varepsilon)}\left(x_T^{\alpha} + \eta \nabla_{x_T^{\alpha}} \mathcal{L}\left(f\left(x_0^{\alpha}\right), c\right)\right). \quad (6)$$

Here, $P_{Bp}$ projects the updated seed onto the $l_p$ ball of radius $\varepsilon$ around $x_T$, with $\eta$ and $\alpha$ representing the learning rate and iteration. However, the gradient of $\mathcal{L}(f(x_0^{\alpha}), c)$ relative to $x_T^{\alpha}$ through the $T$-step diffusion process is computationally infeasible. Therefore, we employ the acceleration technique [7, 81], treating the gradient through the diffusion model as constant, as shown in Equation7:

$$\frac{\partial x_0^{\alpha}}{\partial x_T^{\alpha}} = \frac{\partial x_{T-1}^{\alpha}}{\partial x_T^{\alpha}}\left(\frac{\partial x_{T-2}^{\alpha}}{\partial x_{T-1}^{\alpha}} \cdot \frac{\partial x_{T-3}^{\alpha}}{\partial x_{T-2}^{\alpha}} \cdots \cdot \frac{\partial x_0^{\alpha}}{\partial x_{\alpha}^{\alpha}}\right) \approx k. \quad (7)$$

With this approximation, the back-propagation path can be simplified with only the classifier gradient:

$$\frac{\partial \mathcal{L}\left(f\left(x_0^{\alpha}\right), c\right)}{\partial x_T^{\alpha}} = k \cdot \frac{\partial \mathcal{L}\left(f\left(x_0^{\alpha}\right), c\right)}{\partial x_0^{\alpha}}. \quad (8)$$

In practice, gradient updates like PGD [47] are applied to the denoised $x_0$, transferring the perturbation back to the seed $x_T$. This approach achieves effects similar to EvoSeed with just a few updates. Since perturbations occur at the diffusion seed, they inherently modify high-level features of the denoised results, such as composition and shape. However, as $x_T$ undergoes the whole denoising purification process, perturbing the seed alone may not ensure the desired level of adversarial impact.

**Adversarial Guidance.** Continuous adversarial guidance can be applied throughout generation [6, 10], introducing adversarial perturbations at each denoising step as shown in Equation 9:

$$x_\tau = x_\tau + \xi \cdot \nabla_{x_\tau} \mathcal{L}\left(f\left(x_\tau\right), c\right). \quad (9)$$

Here, $\xi$ determines the scale of perturbations,tailored to the sampling schedule [27]. As adversarial guidance spans from coarse to fine steps, it impacts both high-level and detailed texture features.

**Adversarial Edition.** Further optimization can be performed on the generation result. Integrating a denoising step after each gradient update in PGD can enhance fidelity and transferrability [81]. This is akin to the continuation of adversarial guidance, fine-tuning low-level features until achieving desired adversarial effect.

We sequentially apply three methods, adding adversarial control at various feature granularities. In practice, flexible combinations can be chosen to balance effectiveness and cost.

## 4.2 Watermark Embedding

Unremovability - the robustness against removal attacks [1] ensures that adversaries, even when aware of the underlying watermarking algorithm, should struggle to remove or overwrite the watermark. Model extraction, particularly practical and effective among removal attempts [31, 44], has become a key focus in developing robust watermarking strategies[31, 33, 46].

*4.2.1 The Robustness Pitfall.* Evaluating watermarking algorithms typically involves two aspects: Functionality Preservation and Unremovability. Successful embedding must maintain task performance, as measured by global accuracy metrics, and withstand removal attempts, ensuring that watermark accuracy $\varphi_{wm}$ exceeds

a threshold on the derived surrogate model. Fulfilling both criteria marks the success of a embedding algorithm [3, 31, 33, 46].

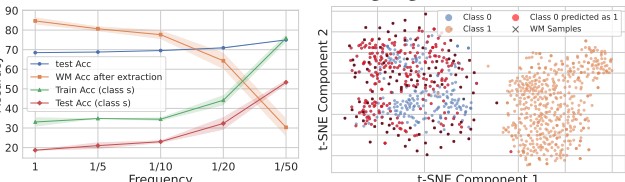

**Figure 2: Generalization and $\varphi_{wm}$ (WM Acc) by Frequency.**  **Figure 3: Feature Space Visualization with t-SNE.**

In this paper, we expose a robustness pitfall in traditional evaluations of embedding algorithms. We present a simplistic algorithm **trivial WM**, which constructs a trigger set by relabeling unmodified samples from a source class $s$ to a target class $t$ and integrates it into the training process at high frequency. Surprisingly, trivial WM achieves high watermark accuracy on extraction surrogates while preserving competitive main task performance. Figure 2 illustrates how trivial WM performs on CIFAR-100 with varying updating frequencies. After training on watermark samples following each standard batch, trivial WM sets the state-of-the-art watermark accuracy on extraction surrogates but significantly undermines generalization in class $s$. Lowering the frequency improves performance on class $s$ but causes a swift decline in watermark accuracy.

Figure 3 delves into this phenomenon by analyzing feature distributions of the watermarked model. High-frequency updates with the trigger set shifts the decision boundary to map the surrounding region of watermark samples as the target class. Although no watermark samples are presented in the query set for extraction, samples close to them in the feature space are misclassified as the target class, smoothly transmitting watermark behavior. In extreme cases, the model learns to label all source class samples as the target class, achieving near-perfect surrogate watermark accuracy, at the cost of a minor 1% decline in overall task performance on CIFAR-100. Thus, sacrificing local decision boundaries for watermark behavior leads to exceptional unremovability. Yet, this is an undesirable outcome, harboring significant risks and acting as a misleading pitfall.

In Section 3, we explored how poisoning-style watermarking introduces evasion-prone shortcuts. The robustness pitfall reveals a deeper issue: watermark persistence necessitates decision boundary perturbation driven by misclassifications, creating a conflict between evasion and watermark robustness. The unremovability of current watermarks cannot be decoupled from adverse effects.

*4.2.2 Learning a Friendly Teacher.* UAE watermark is naturally entangled with the task distribution, conceptually easy to survive in extraction. However, adversaries face optimization challenges in transferring knowledge via ERM [68]. Although the surrogate exhibit even better generalization performance, it does not faithfully mimic all behaviors of the watermarked model [18, 48, 50, 68]. Since the trigger set is absent from extraction queries [1], transferring watermark behavior is challenging [31, 52]. Poisoning-style watermarks compensate by exploiting the robustness pitfall phenomenon, but inevitably introduces evasion vulnerabilities.

Therefore, we propose a novel embedding strategy: learning the watermarked model as a "friendly teacher" adept at sharing knowledge, guiding the surrogate to learn watermark behavior from limited queries. Training a better teacher is an underexplored

direction, with current methods focusing on collaborative training between teacher and student networks [57, 63]. However, model owners lack control over the surrogates' training process. Thus, we attempt to learn a surrogate-agnostic friendly teacher from the perspectives of function mapping and output distribution properties.

**Function Mapping Properties.** Recent studies theoretically indicate that teacher models should exhibit Lipschitz continuity and transformation equivariance, making them easier to emulate [14]. Lipschitz continuity implies the model $f$ reacts minimally to slight input changes, i.e., $||f(x) - f(x')|| \leq L||x - x'||$, with $|| \cdot ||$ representing a distance metric. However, exact computation of the Lipschitz constant $L$ is notoriously difficult [2]. Yet, adversarial robustness ties closely to the Lipschitz constant [16, 88]. Employing UAEs as the trigger set naturally promotes local Lipschitz smoothness around watermark samples. We enhance this by varying data augmentation strategies for watermark samples per batch [49].

For transformation equivariance, we leverage consistency regularization [71], and choose random erasing as a highly controllable transformation [87]. It replaces small patches from the original image with random colors. The model $f$ is encouraged to produce similar outputs for the transformed and original samples:
$$\mathcal{L}_{cr} = KL(f(x)||f(re(x))). \tag{10}$$
Here, $re(\cdot)$ is the random erasing operator, and $KL(\cdot||\cdot)$ calculates the Kullback-Leibler divergence. In practice, $\mathcal{L}_{cr}$ serves as a regularization term, jointly optimized with the classification loss.

**Output Distribution Properties.** Extraction utilize soft labels from query responses, leveraging their rich information for efficient mimic [52]. However, modern networks often exhibit overconfidence [22, 50], which obstructs knowledge transfer. To mitigate this, temperature scaling is employed in knowledge distillation $\Gamma$ to soften softmax layer outputs for both teacher and student, with $p_i = \sigma(z_i/\Gamma)$, where $z_i$ represents logits and $\sigma$ the softmax function. However, model owners lack control over the evasion adversary, and temperature is often ignored in extraction ($\Gamma = 1$) [3, 31, 33, 52, 75]. We propose to apply temperature scaling to protected model, regardless of the extraction settings. As a result, the loss function of black-box extraction becomes Equation 11:
$$\mathcal{L}_{EXT} = -\Gamma^2 \sum_{i=1}^{K} \sigma_i \left( \frac{z_v}{\Gamma \cdot \Gamma_v} \right) \log \sigma_i \left( \frac{z_u}{\Gamma} \right). \tag{11}$$
where $z_v$ and $z_u$ are the output logits of the protected model and extraction surrogate, respectively. $\Gamma$ is the adversary's distillation temperature chosen during extraction attempts, while $\Gamma_v$, set by the model owner at the API level, adjusts the output distribution. The adversary, limited to black-box API output access, remains unaware of $\Gamma_v$. Incorporating $\Gamma_v$ modifies the gradient of the extraction loss for the $j$th class as detailed in Equation 12:
$$\nabla_{z_{s_j}} \mathcal{L}_{EXT} = -\Gamma^2 \sum_{i=1}^{K} \left[ \sigma_i \left( \frac{z_v}{\Gamma \cdot \Gamma_v} \right) \left( \delta_{ij} - \sigma_j \left( \frac{z_u}{\Gamma} \right) \right) \right]. \tag{12}$$
where $\delta_{ij}$ is the Kronecker delta function (1 for $i = j$, 0 otherwise), $\Gamma_v$ reduces class discrepancies of the protected model's output distribution, guiding gradient updates to align logits beyond focusing solely on the single correct class.

To preserve watermark accuracy despite minor parameter adjustments, we further seek local optima with parameter neighbourhood continuing to exhibit the watermark behavior. This is achieved by

bi-level optimization, minimizing watermark loss under the worst-case conditions within the parameter vicinity:
$$\min_{\theta} \max_{\|\delta\|_p \leq \varepsilon} \mathcal{L}(f_{\theta+\delta}(x), y). \tag{13}$$
The inner optimization seeks the worst weight perturbation $\delta$ within the $p$-norm bound to remove the watermark, while the outer optimization adjusts the parameters to preserve watermark memorization. We employ an approximation in the form of sharpness-aware minimization [17] to efficiently solve the inner problem:
$$\hat{\delta} \approx \varepsilon \cdot \frac{\nabla_\theta \mathcal{L}(f_\theta(x), y)}{\|\nabla_\theta \mathcal{L}(f_\theta(x), y)\|}. \tag{14}$$
Intuitively, sharpness-aware minimization aims for solutions with flatter loss landscapes to prevent adversaries from easily shifting parameters that support watermark behavior. Unlike traditional watermarking [3, 31, 33, 46], our training strategies apply to the entire training set, not just the trigger set. Our objective is to learn the protected model with desirable properties for watermarking, rather than sacrificing the main task to focus on the watermark.

*4.2.3 Watermark Verification.* Our verification approach is similar to traditional trigger-set watermarks, assessing ownership by evaluating watermark accuracy $\varphi_{wm}$. Yet, $\varphi_{wm}$ overlooks how third-party models perform on the watermark samples, potentially causing false alarms. Therefore, we introduce a self-calibration method: select control samples from the generated UAE set, matching the trigger set in size, which can mislead the model post-watermarking. The watermark samples and the control samples represent the *pros* and *cons* of the watermarked model. We use the difference in accuracy ($\varphi_{pros} - \varphi_{cons}$) as a similarity metric between the suspect and protected models, akin to $\varphi_{wm}$.

## 5 Experiments
In this section, we evaluate the performance of UAE watermarking against leading methods in terms of evasion and watermark robustness. Comparative methods include the pattern-based approaches EWE from USENIX Security 21'[31] and RS from ICML 22' [3], as well as pattern-free approaches MBW from ICML 23' [33] and MEA from S&P 24'[46]. Experiments are conducted on the CIFAR-10/100 datasets [3, 31, 33, 36, 46], standard benchmarks for model watermarking, and the more challenging high-resolution Imagenette dataset, a 10-class subset of Imagenet [12]. We use ResNet-18 [24], the largest-scale model employed by comparative methods [3, 31, 46], for consistent performance evaluation. Exploration of more advanced structures is discussed in Section 5.2.2. Code and detailed settings are available at ⚙.

### 5.1 Robustness against Evasion Adversaries
Evaluations of evasion robustness are conducted on the Imagenette dataset to better mirror real-world conditions. We instantiate Equation 1 with the Pixel Backdoor [74] and implement its untargeted universal attack form as described in Equation 2. Instance-specific adversarial attacks employ the $L_0$ constraint AutoAttack [9, 86]. The perturbation pixel limits for Pixel backdoor and AutoAttack are set at 200 and 50, respectively, accounting for less than 0.5% and 0.1% of the original image pixels.

*5.1.1 Overall Evaluation.* We conduct a coarse-grained assessment through untargeted attack against all watermarking algorithms and normal models without watermark, as shown in Table 1.

**Table 1: Evasion Robustness Against Untargeted Attacks (Realtive means relative ASR compare with normal models).**

| Method | Clean(ACC)↑ | PixelBackdoor(ASR)↓ | | SparseAuto(ASR)↓ | |
|---|---|---|---|---|---|
| | avg&std | avg&std | relative | avg&std | relative |
| Normal | 98.32±0.54 | 2.44±0.80 | - | 32.36±1.76 | - |
| EWE | 95.92±0.64 | 83.20±4.55 | 80.76 | 84.76±5.04 | 52.40 |
| RS | 97.24±0.17 | 71.04±11.24 | 68.60 | 72.48±12.76 | 40.12 |
| MBW | 89.80±1.07 | 12.28±1.37 | 9.84 | 38.88±0.66 | 6.52 |
| MEA | 87.92±1.66 | 67.08±11.63 | 64.64 | 80.08±3.88 | 47.72 |
| UAE | 98.00±0.20 | 2.52±0.36 | **0.08** | 21.20±0.93 | **-11.16** |

Evasions achieve only modest attack success rate (ASR) on models without watermark due to the lack of shortcuts. Pattern-based watermarking slightly decreases generalization by about 2%, but the shortcuts introduced significantly amplify vulnerabilities. Under the same budget, ASR of Pixel Backdoor increases by about 30 times, and that of AutoAttack by over 40%. In contrast, pattern-free watermarking induces a 10% drop in generalization performance. Even without trigger patterns, vulnerability of MEA to evasion is comparable to pattern-based watermarking. Trigger inversion can exploit shortcuts without matching the exact pattern, and abandoning patterns does not lessen the risks. Among traditional watermarks, only MBW exhibits a relatively minor decrease in robustness. However, this does not imply the harm is negligible, with further details in Section 5.1.3. Finally, UAE watermarking matches the generalization performance of unwatermarked models, further reducing ASR of AutoAttack by 11.16%. The knowledge injection of UAE watermarking teaches models to recognize hard samples, avoiding the creation of shortcuts while patching inherent vulnerabilities, thereby enhancing their adaptability to challenging scenarios.

| Method | PixelBackdoor(ASR) | |
|---|---|---|
| | avg&std | relative |
| Normal | 8.40±3.58 | - |
| EWE | 80.80±5.76 | 71.67 |
| RS | 75.20±7.43 | 66.07 |

**Table 2: Attack Success Rate from Source Class $s$ to Target Class $t$.**

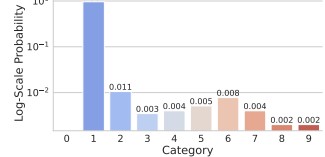

**Figure 4: Distribution of Target Classes for Attacked Samples.**

*5.1.2 Class-wise Evaluation of Backdoor Watermarking.* We explore the link between decrease in robustness and shortcuts in pattern-based watermarking, which uses patterns to flip source class $s$ samples to target classes $t$. Therefore, we utilize Pixel Backdoor to solve Equation 1 and compare the targeted ASR from $s$ to target class $t$ for EWE, RS and unwatermarked models in Table 2.

In this scenario, trigger inversion directly attempts to reconstruct the patterns, resulting in significantly higher ASRs compared to unwatermarked models. Furthermore, Figure 4 visualizes the target class distribution for samples successfully attacked with untargeted Pixel Backdoor on EWE models. Although the attack indiscriminately targets all classes, the target classes used for watermark (class 1) become vulnerable entry points. Samples from various classes are easily perturbed to be classified as the target class, posing substantial risks in security-related scenarios.

*5.1.3 In-depth Analysis of Noise Label Watermarking.* MBW employs adversarial training (AT) on its label noise trigger set to increase *margin*. In Figure 5, we visualize the robust accuracy under a 5-step $L_\infty$ PGD attack for MBW, MBW without *margin*, MBW with AT on the training set, and an unwatermarked model with AT.

AT on the trigger set enhances the robustness of MBW, influencing fair comparison in Section 5.1.1. When both the MBW and normal models undergo AT, there consistently exists a gap in adversarial robustness. AT cannot fully compensate for the vulnerability introduced by watermark embedding. In Figure 6, we compare the attack effects of patterns derived from MBW using noise label trigger inversion with those of Gaussian noise. Effectiveness of the recovered pattern far surpasses that of Gaussian noise, explicitly demonstrating the existence of shortcuts. As attacks evolve, these shortcuts will continue to be exploited.

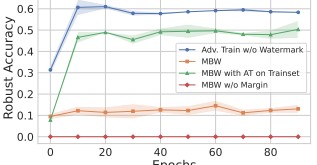 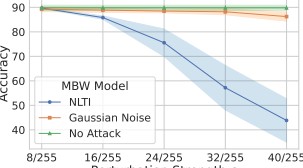

**Figure 5: Robustness Accuracy in Different Epochs.** | **Figure 6: Robustness Accuracy under Different Perturbations.**

## 5.2 Robustness against Stealing Adversaries

This section evaluates watermarks in resisting removal attacks. For *model modification*, we iteratively prune and fine-tune the model [43] on 20% of the training set until validation accuracy drops more than 5% or pruning rate exceeds 75%. For *model extraction*, we employ Knockoff Nets [52], using the training set as queries [31, 33, 46]. For *input preprocessing*, we employ CLIP [59] to extract features and then use 10% of the training set to create isolation forests [42] for each class, filtering out potential watermark samples. To reduce overhead, we employ only the first 2 blocks of the feature extractor for CIFAR-10/100 and the first 3 blocks for Imagenette.

Table 3 presents the generalization performance and watermark accuracy of all methods. UAE watermarking achieves the highest main task accuracy across all datasets. We calculate watermark accuracy ($\varphi_{wm} = \varphi_{pros} - \varphi_{cons}$) as the difference between accuracy of trigger set and the UAE control group. This method yields $\varphi_{wm}$ values of -1%, 0.2% and 0% for unwatermarked models on three datasets, confirming the effectiveness of self-calibration. For removal attacks, UAE watermarking exhibits the best average robustness. Notably, without relying on robustness pitfalls, UAE watermarking achieves the highest $\varphi_{wm}$ on extraction surrogates, illustrating the advantages of its "friendly teacher" learning approach. Furthermore, since UAEs adhere to the original distribution, they are difficult to identify in anomaly detection. Although UAE watermarking is not always the most robust against fine-pruning, modifications have never reduced $\varphi_{wm}$ below 85% without significantly impacting generalization. In cases of the same validation accuracy drop, both MBW and MEA tolerate higher pruning rates, with MBW even showing improved generalization on CIFAR-10, cause they adapt to watermarks by significantly sacrificing representation capacity.

*5.2.1 Analysis of the Robustness Pitfall.* Pattern-free watermarking exhibit competitiveness in extraction. However, their unremovability mainly stems from the robustness pitfall. On CIFAR-10, MBW and MEA achieve only 88.46% and 95.31% **training accuracy**, respectively, while a normally trained Resnet-18 approaches perfect accuracy. As the extraction queries reuse the training set, we filter out all misclassified samples and repeat experiments, resulting in $\varphi_{wm}$ on extraction surrogates of MBW and MEA dropping to 53.4%

**Table 3: Comparative Analysis of Generalization and Watermark Accuracy Across CIFAR-10/100 and Imagenette on Watermark Model and Removal Attacks.**

| Dataset | Method | Victim | | Fine-pruning | | Extraction | | Anomaly Detection | | Avg Acc on |
|---|---|---|---|---|---|---|---|---|---|---|
| | | Main Task Acc | Trigger Set Acc | Main Task Acc | Trigger Set Acc | Main Task Acc | Trigger Set Acc | Main Task Acc | Trigger Set Acc | Trigger Set |
| CIFAR10 | EWE | 93.26±0.34 | 99.80±0.45 | 89.62±0.20 | 81.40±4.67 | 93.91±0.24 | 40.80±22.07 | 88.22±0.32 | 78.80±7.01 | 67.00 |
| | RS | 93.95±0.22 | 100.00±0.00 | 90.62±0.73 | 75.80±12.77 | 94.64±0.20 | 2.40±0.55 | 88.59±0.23 | 73.80±1.79 | 50.67 |
| | MBW | 85.73±2.38 | 99.80±0.45 | 87.19±0.28 | 10.40±1.82 | 89.00±2.17 | 78.60±5.27 | 80.85±2.52 | 75.20±2.49 | 54.73 |
| | MEA | 85.03±1.51 | 99.00±0.71 | 81.22±0.97 | 79.40±7.09 | 89.38±0.82 | 87.80±4.21 | 80.23±1.59 | 34.20±2.49 | 67.13 |
| | UAE | 94.04±0.11 | 100.00±0.00 | 90.13±0.38 | 87.40±4.28 | 94.59±0.15 | 88.00±3.16 | 88.82±0.22 | 91.80±1.48 | 89.07 |
| CIFAR100 | EWE | 72.75±0.06 | 100.00±0.00 | 68.63±0.87 | 91.80±3.03 | 75.29±0.22 | 16.00±2.74 | 65.36±0.39 | 17.40±8.93 | 41.73 |
| | RS | 78.84±0.52 | 100.00±0.00 | 75.43±1.06 | 64.40±23.64 | 77.48±0.67 | 3.20±0.84 | 71.25±0.57 | 47.60±12.16 | 38.40 |
| | MBW | 69.22±1.40 | 100.00±0.00 | 66.24±1.49 | 91.00±5.34 | 75.36±1.00 | 52.40±8.20 | 62.64±1.17 | 62.80±5.72 | 68.73 |
| | MEA | 59.11±3.77 | 99.60±0.55 | 56.28±2.20 | 99.20±0.84 | 64.34±3.47 | 71.80±16.71 | 52.58±3.62 | 37.80±6.10 | 69.60 |
| | UAE | 79.60±0.95 | 100.00±0.00 | 75.14±0.61 | 96.60±2.07 | 79.05±0.83 | 83.00±2.45 | 71.64±1.08 | 94.00±1.22 | 91.20 |
| IMAGENETTE | EWE | 95.92±0.64 | 100.00±0.00 | 93.12±0.78 | 35.40±8.85 | 96.44±0.26 | 15.80±5.07 | 90.76±0.55 | 61.40±3.85 | 37.53 |
| | RS | 97.24±0.17 | 100.00±0.00 | 94.04±0.38 | 60.20±23.31 | 96.40±0.40 | 5.80±5.36 | 91.66±0.55 | 52.80±5.40 | 39.60 |
| | MBW | 89.80±1.07 | 100.00±0.00 | 85.72±0.66 | 53.20±21.21 | 91.84±0.71 | 33.20±5.40 | 85.84±1.34 | 36.00±4.00 | 40.80 |
| | MEA | 87.92±1.66 | 100.00±0.00 | 83.28±1.80 | 97.40±1.52 | 92.08±1.40 | 49.28±28.40 | 84.68±1.55 | 1.20±1.79 | 49.29 |
| | UAE | 98.00±0.20 | 100.00±0.00 | 94.32±0.50 | 92.20±6.10 | 97.56±0.22 | 68.00±10.84 | 93.24±1.09 | 93.40±2.30 | 84.53 |

and 30.4%. Figure 7 visualizes the feature distribution for the source and target classes of MEA and an unwatermarked model. Similar to **Trivial WM**, MEA induces misclassifications on the queries, easily transferring watermark behavior. In contrast, UAE watermarking achieves 100% training accuracy on CIFAR-10, without relying on any robustness pitfall phenomenon.

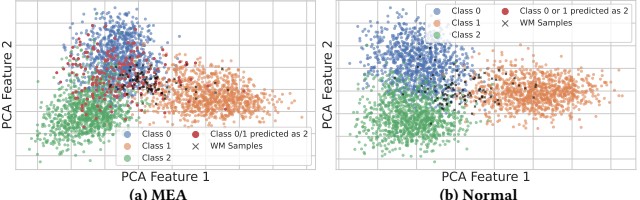

(a) MEA  (b) Normal
**Figure 7: Feature Space Visualization for MEA and normal models.**

The robustness pitfall connects poisoning-style watermarking to its adverse effects: the stronger the watermark, the more pronounced the vulnerabilities. Patterns recovered using the MEA extraction surrogate yield a 47.88% transfer ASR on MEA watermarked model, while those from a normal model achieve only a 15.88% transfer ASR. Adversaries targeting evasion rather than stealing even expect the extraction surrogates learn the watermark behavior better, facilitating successful evasions.

**Table 4: Performance Comparison of Watermarking Methods Using OOD Samples as Query Sets for Model Extraction.**

| Method | CIFAR 10 | | CIFAR 100 | |
|---|---|---|---|---|
| | Main Task Acc | Trigger Set Acc | Main Task Acc | Trigger Set Acc |
| MEA | 83.20±1.16 | 90.67±1.53 | 50.18±6.89 | 64.00±22.34 |
| MBW | 82.98±0.87 | 76.00±7.00 | 58.42±3.00 | 65.33±1.53 |
| UAE | 93.06±0.10 | 97.00±1.00 | 74.05±0.19 | 94.67±1.15 |

*5.2.2 Advanced Extraction Scenario.* In this section, we explore complex extraction scenarios. Table 4 shows watermark performance using out-of-distribution (OOD) samples as queries: CIFAR-10 watermarked model use CIFAR-100 as queries, and vice versa. $\varphi_{wm}$ on UAE watermarking surrogates is even higher than in-distribution sample extraction, as separating the queries from training set facilitates knowledge transfer. Poisoning-style watermarks often struggle with large-scale networks [31, 46]. We conduct experiments on CIFAR-10 using EfficientNetV2 [72], with both same-architecture models and Resnet-18 serving as extraction surrogates, as shown in Table 5. UAE watermarking retains satisfactory unremovability in modern networks and cross-architecture extraction.

**Table 5: Comparison of Extraction Results Using Different Models.**

| Surrogate | Victim | | Extraction | |
|---|---|---|---|---|
| | Main Task Acc | Trigger Set Acc | Main Task Acc | Trigger Set Acc |
| efficientnet v2 | 95.10±0.14 | 100.00±0.00 | 95.41±0.09 | 81.00±1.00 |
| resnet18 | | | 94.91±0.05 | 80.00±3.61 |

*5.2.3 Adaptive Removal Attack.* Since trigger sets in UAE watermarking inherently contain adversarial examples, we further investigated whether typical adversarial defenses hinder watermark verification. Table 6 shows the impact of adversarial fine-tuning. While $\varphi_{wm}$ of UAE decreases, it still outperforms MBA and MEA, indicating adversarial fine-tuning does not expose specific vulnerabilities of UAE watermarking. Figure 8 depicts the removal results of randomized smoothing [3], which affects UAEs far less than regular adversarial samples. The flexibility of unbounded adversarial samples to circumvent defenses makes them particularly effective for constructing resilient trigger sets, adapting to the diverse stealing adversaries in the open world.

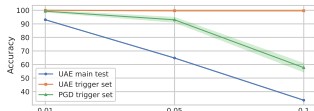

**Figure 8: Impact of Randomized Smoothing.**

**Table 6: Impact of Adversarial Fine-tuning.**

| Method | Adv. finetuning | |
|---|---|---|
| | Main Task Acc | Trigger Set Acc |
| MBW | 82.29±0.62 | 25.00±2.65 |
| MEA | 80.43±0.60 | 32.00±2.00 |
| UAE | 88.93±0.95 | 55.67±9.71 |

## 6 Conclusion

In this paper, we identify the dilemma that poisoning-style model watermarks increase susceptibility to evasion while protecting against theft. To tackle this issue, we introduced a novel, reliable watermarking algorithm utilizing knowledge injection as unique identifiers and optimizing knowledge transfer to enhance watermark behaviors. Experimental results demonstrate that our UAE watermarking not only outperforms SOTA methods in unremovability but also avoids evasion exploitation. This dual effectiveness underscores its potential as a comprehensive solution to protect deep learning models from a spectrum of complex threats.

## 7 Acknowledgements

This work was supported by the Natural Science Foundation of Jiangsu Province of China (BK20241272), the Fundamental Research Funds for the Central Universities (2242024k30059), and the Start-Up Research Fund of Southeast University (RF1028623129).

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
