# OpenReview forum: "Reliable Model Watermarking: Defending Against Theft without Compromising on Evasion"
_acmmm.org/ACMMM/2024/Conference — MM2024 Poster_

### Official Review · Reviewer_t2c4 · 2024-05-23

**Rating:** 4
**Confidence:** 3

**Summary:**

This paper identifies that traditional poisoning-style watermarking introduces exploitable shortcuts, making models more susceptible to evasion attacks. To mitigate this, the authors propose a novel watermarking method using diffusion models to generate Unrestricted Adversarial Examples (UAEs) as trigger sets. This approach focuses on knowledge injection rather than error memorization.

**Strengths:**

1. Innovative Use of Diffusion Models
2. Comprehensive Robustness
3. Extensive Experimental Validation

**Limitations:**

1. While the method shows promising results on the tested model (ResNet), its effectiveness on a wider range of model architectures and more complex datasets needs further validation.
2. It is unclear whether the robustness mainly stems from UAE, rather than from the proposed method itself.
3. In Table 1, why is UAE's SparseAuto (ASR) relative value negative (-11.16)? If I understand correctly, a reasonable value should be around 0.
4. I am curious about the false positive (FP) results. Would this method falsely detect other original models as watermarked? Is there any guarantee for FP, similar to the approach discussed in https://arxiv.org/abs/2305.03807?

**Suitability:**

2

---

### Official Review · Reviewer_6dVS · 2024-05-24

**Rating:** 4
**Confidence:** 3

**Summary:**

The paper introduces a model watermarking technique to protect deep learning models from theft without compromising their ability to evade adversarial attacks. It tries to address the shortcomings in current trigger-set watermarking methods that make models vulnerable to evasion by exploiting shortcuts created during the watermark embedding process. Their proposed solution uses diffusion models to generate unrestricted adversarial examples as trigger sets and optimizes knowledge transfer to teach models to identify these examples accurately, thus enhancing resistance to both evasion and removal attacks. From the experimental results on datasets like CIFAR-10/100 and ImageNet, the model's robustness and effectiveness are improved against state-of-the-art solutions.

**Strengths:**

1. The novel approach for a necessary research question.
The paper presents a novel approach to an important research question in the field of machine learning security, that is, how to protect intellectual property in deep learning models while maintaining their robustness against adversarial attacks. The necessity of this research question arises from the increasing adoption of Machine Learning as a Service (MLaaS) platforms while democratizing access to sophisticated models and exposing them to theft and unauthorized use.

2. Comprehensive experiment evaluation.
This paper thoroughly evaluates the proposed method against diverse evasion and removal attack scenarios based on multiple datasets, CIFAR-10/100 and ImageNet.

**Limitations:**

1. Some sentences are unclear, and the writing needs improvement.
For instance, in the introduction, “we identify that all poisoning-style watermarks, even those crafted with random label noise trigger sets, embed shortcuts into the protected model” should be replaced with “we identify that all poisoning-style watermarks, even those crafted with random label noise trigger sets and embed shortcuts into the protected model”. Similarly, in this paper, many sentences are challenging to understand, which should be further polished.

2.Some details can be supplemented.
First, in the introduction, the current technology challenges need to be clarified directly, and this can illustrate the advancement of this paper more clearly.
Second, although this paper aims to make watermarks hard to remove by focusing on knowledge transfer, the adaptive attackers who can learn this transfer might discover ways to circumvent or remove the watermarks.
Third, in Section 5.2.2, the out-of-distribution (OOD) question is considered on CIFAR-10 and CIFAR-100 datasets, while similar data is not the best choice for researching this problem.
Fourth, the efficiency of the proposed method was not evaluated, and it is curious whether complex operations bring more computational overhead to the model.
These problems require clarification from the authors.

3. The code or its implementation can be further supplemented. In this way, researchers can reproduce the solution and understand its details better.

**Suitability:**

2

---

### Official Review · Reviewer_uwsT · 2024-05-25

**Rating:** 4
**Confidence:** 3

**Summary:**

This paper proposes to generate Unrestricted Adversarial Examples (UAEs) for watermark embedding via diffusion models, enhancing robustness against removal attacks. The proposed method leverages knowledge injection during training to maintain watermark properties without degrading model performance. The experimental results on CIFAR-10 and Imagenette datasets demonstrate that this method significantly improves watermark resilience and model performance in adversarial settings.

**Strengths:**

1.	The research topic of defending against model stealing is interesting and very important. This paper proposes a novel scheme by utilizing diffusion models to synthesize UAEs as the trigger set.
2.	The experiments are also comprehensive to some extent. The paper includes detailed ablation studies to validate the effectiveness of the proposed method.

**Limitations:**

1.	There are some harmless watermark methods (e.g., [1-3]) and model fingerprints (e.g., [4-5]) that should be used as baselines for comparison in the paper.
2.	More detailed information about the diffusion model (DM) is needed. This method requires an external diffusion model (DM), but the experimental setup in the main text does not seem to describe this DM in detail (e.g., structure, training dataset). I am particularly curious if the training dataset for this DM needs to be similar to the training dataset of the protected model. If their training datasets differ significantly in distribution, is the proposed method still effective?
3.	Increase the font size of the title in Figure 1. Ensure consistency in the use of uppercase and lowercase letters in the figures, such as "evasion-adversary" being lowercase while other parts are uppercase.
4.	All tables should highlight the best-performing data in bold.
References:
[1] Defending against model stealing via verifying embedded external features. AAAI 2022.
[2] Move: Effective and harmless ownership verification via embedded external features. 2022.
[3] Free fine-tuning: A plug-and-play watermarking scheme for deep neural networks. ACM MM 2023.
[4] Fingerprinting deep neural networks globally via universal adversarial perturbations. CVPR 2022.
[5] Metafinger: Fingerprinting the deep neural networks with metatraining. IJCAI 2023.

**Suitability:**

2

---

### Meta-Review · Area_Chair_ZE2R · 2024-06-28

**Recommendation:** Accept (Poster)
**Confidence:** 5

**Metareview:**

In this paper, the authors introduce a novel model watermarking technique to protect deep learning models from theft without compromising their ability to evade adversarial attacks. It leverages knowledge injection during training to maintain watermark properties without degrading model performance. Evaluations against several stealing attacks demonstrate the effectiveness of the method. Reviewers generally consider the method novel and effective, and the paper is well written. Although some issues were raised, they were well addressed during rebuttal. All reviewers support acceptance.